# ClsVC: Learning Speech Representations with two different classification tasks.

## Abstract

Voice conversion(VC) aims to convert one speaker's voice to generate a new speech as it is said by another speaker. Previous works focus on learning latent representation by applying two different encoders to learn content information and timbre information from the input speech respectively. However, whether they apply a bottleneck network or vector quantify technology, it is very difficult to perfectly separate the speaker and the content information from a speech signal. In this paper, we propose a novel voice conversion framework, 'ClsVC', to address this problem. It uses only one encoder to get both timbre and content information by dividing the latent space. Besides, some constraints are proposed to ensure the different part of latent space only contains separating content and timbre information respectively. We have shown the necessity to set these constraints, and we also experimentally prove that even if we change the division proportion of latent space, the content and timbre information will be always well separated. Experiments on the VCTK dataset show ClsVC is a state-of-the-art framework in terms of the naturalness and similarity of converted speech.

## 1 Introduction

Voice conversion (VC) is an exciting topic committed to converting one utterance of a source speaker into another utterance of a target person by keeping the content in the original utterance while replacing it with the vocal features from the target speaker. Up to now, many methods have been applied in VC successfully. Commonly, these methods can be roughly categorized into two classes, i.e., parallel VC and non-parallel VC Mohammadi & Kain (2017). Specifically, parallel VC means that model training requires parallel corpus, which is unnecessary for non-parallel VC. Recently, more researchers have focused on the solutions of non-parallel VC since it is not easy for us to collect so many paired source-target speech datasets.

Early VC systems, like Gaussian Mixture Model (GMM), Stylianou et al. (1998); Toda et al. (2007) needed a lot of parallel data for model training, and the generated speech quality was not good enough. With the advance of deep learning, a variety of novel VC methods have been proposed in recent years. Among them, GAN-based model is one of the most popular methods, Goodfellow et al. (2014); Hsu et al. (2017a); Kaneko & Kameoka (2018); Kaneko et al. (2019a); Kameoka et al. (2018); Kaneko et al. (2019b) which could learn a global generative distribution of the target speech without explicit approximation. These GAN-based models jointly train a generator and a discriminator. An adversarial loss derived from the discriminator is used to encourage the generator outputs to build indistinguishable from real speech. Thanks to the cycle consistency training, GAN-based VC models can be trained with non-parallel speech datasets.

Besides, learning discrete speech representations has also attracted much attention. Vector Quantization (VQ), an extremely important signal compression method, which can quantify continuous data into discrete data. Previous studies have confirmed that the quantized discrete data produced by the continuous speech data is closely related with the phoneme information Chorowski et al. (2019). Recently, VQVC Wu & Lee (2020) has been proposed to learn to disentangle the content and speaker information with reconstruction loss only. Then, VQVC+ Wu et al. (2020) was soon presented to improve the conversion performance of VQVC by adding the U-Net architecture within an auto-encoder-based VC system. To far improve the performance of disentangling content and speaker information, many other existing studies were introduced to combine with VQ, such as

VQ-Wav2Vec, VQ-VAE and VQ-CPC. Baevski et al. (2019); Ding & Gutierrez-Osuna (2019); van Niekerk et al. (2020)

There is also another line of research focus on learning latent representations with Autoencoder. In particular, Variational Auto Encoder(VAE) is the most famous. The network structure of VAE contains an encoder and a decoder and the core idea is very clear: the encoder learns a specific latent space from input speech and the decoder outputs a reconstructed speech from this latent space. In this process, VAE focuses on how to force the encoder to learn a specific latent space. So far, many VAE-based models have been successfully applied to VC Hsu et al. (2016); van den Oord et al. (2017); Hsu et al. (2017b); Ding & Gutierrez-Osuna (2019); Hsu et al. (2017a). In addition, AutoVC is another successful application of Autoencoder. Qian et al. (2019) Through ingenious experimental design, AutoVC uses two different encoders to learn content and speaker information, respectively, so that this model can achieve distribution-matching style transfer by only on a self-reconstruction loss.

Unfortunately, in the field of VC, all the models mentioned above have their inherent disadvantages. For example, GAN-based models can usually achieve a good conversion effect and ensure the matching between the generated data and the input data, but it is recognized that the training of GAN is very unstable. On the contrary, the training of VQVC is simple and fast enough, but the quantity of audio produced by this method is very poor. This may be because the discrete speech representations will inevitably lose some content information. In addition, although the VAE-based model also has a great conversion effect, it can not guarantee distribution matching. AutoVC is a great study, the training is very simple and achieves state-of-the-art results. However, in order to realize style conversion, it has to introduce a pre-trained speaker encoder.

Based on these existing methods, we naturally wonder if there is a new solution that can achieve the distribution matching as AutoVC and GAN, that trains as easily as VQVC and VAE, that can disentangle content and speaker information by only one encoder as VQ does, and also has better performance in voice conversion or in decoupling linguistic and timbre information from speech?

In this paper, we proposed a novel voice conversion framework to meet all the above requirements. Specifically, our model is similar to VAE, Autoencoder is the main framework of our model, and two different types of classification tasks are applied to force our model to separate the content and speaker information correctly. Here, the two classification tasks refer to general classification tasks and adversarial classification tasks respectively. The goal of the general classification task is to identify the features related to the speaker as accurately as possible, that is, the speaker information. While the latter is designed to eliminate speaker information in latent space to get speaker-independent features, that is, content information. Experiment results are carried out on the VCTK dataset. Objective and Subjective evaluations demonstrate that the proposed method outperforms VQVC, AutoVC, VQ-VAE and StarGAN-VC in terms of naturalness and speaker similarity.

## 2 BACKGROUND

In mathematical statistics, if we already know the joint probability density functions of $X$ and $Y$, we can easily find the marginal probability density functions of $X$ and $Y$ respectively. Formally, if $(x, y) \sim p(x, y)$ is known, we can get the marginal distributions of $X$ and $Y$ by the following formula:

$$f_X(x) = \int p(x, y)d\boldsymbol{y} \qquad\qquad f_Y(y) = \int p(x, y)d\boldsymbol{x} \qquad\qquad (1)$$

Further, under the constraints of some setting conditions, although the closed-form of joint probability density functions $p(x, y)$ in Eq. (1) is generally unknown, it is still feasible for a neural network to learn the marginal distributions of $x$ and $y$ from input samples $z$ when each $z$ corresponds to a unique pair of $(x, y)$.

**Mutual information (MI)**, a crucial indicator to measure the dependence between two different variables. Recently, many MI estimators have been successfully applied to constrain neural networks to disentangle different components of the input data. which can be formulated as

$$\mathcal{I}(X, Y) = \int_X \int_Y P(X, Y) \log \frac{P(X, Y)}{P(X)P(Y)} \qquad\qquad (2)$$

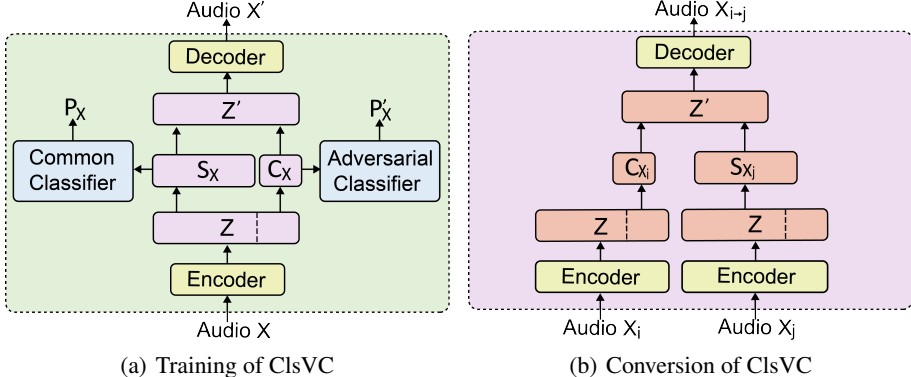

(a) Training of ClsVC   (b) Conversion of ClsVC

Figure 1: Framework of ClsVC. $Z$ is the latent variable, which is divided into two parts, namely $C_X$ and $S_X$. Here, we assume that $C_X$ represents content information, which is speaker independent, and $S_X$ represents speaker information, which is closely related to speaker identity.

Where $P(X)$ and $P(Y)$ are the marginal distributions of $X$ and $Y$ respectively, and $P(X, Y)$ denotes the joint distribution of $X$ and $Y$. Since it is hard to obtain the required distribution formula $P(X, Y)$, many studies focus on proposing a sample-based MI lower bound (or upper bound) to get an approximation that can be calculated. Moon & Hero (2014); Hjelm et al. (2018); Cheng et al. (2020)

Recently, Yuan *et al.* Yuan et al. (2020) , introduced a new MI estimator to learn content and style information from speech for voice conversion. Specifically, they proposed a novel MI-based learning objective to encourage a content encoder to output the content embedding and guide a speaker encoder to output the speaker embedding. Inspired by this, we proposed a new, simple and more effective framework for learning latent speech representation.

**Gradient Reversal Layer (GRL)** Gradient Reversal Layer (GRL) was first proposed to address the issue of domain adaption Ganin & Lempitsky (2015), which aims to force model to output domain-shared features which are independent with domains. Specifically, GRL is often located between an encoder and a domain classifier. During the forward propagation, GRL acts as an identity transform. During the backpropagation, GRL takes the gradient from the subsequent level, multiplies it by $-1$ and passes it to the preceding layer so that the encoder and domain classifier have completely opposite optimization objectives.

## 3 METHOD

Firstly, for every speech $\boldsymbol{x}$ , we use content embedding $\boldsymbol{C_x}$ to represent linguistic information and speaker embedding $\boldsymbol{S_x}$ is proposed to represent timbre and style information. And, $\boldsymbol{U}$ means the set of speakers. The following two theorems are the premise of our framework:

**Theorem 3.1** *The content embedding $\boldsymbol{C_x}$ and speaker identity $\boldsymbol{U}$ are independent for each other, In addition, the probability of each speaker's speech being selected is the same, Formally, $P(\boldsymbol{U} = \boldsymbol{u}|\boldsymbol{C_x}) = P(\boldsymbol{U} = \boldsymbol{u}) = \boldsymbol{constant}$ regardless of the speaker identity $\boldsymbol{u}$ .*

**Theorem 3.2** *The speaker embedding $\boldsymbol{S_x}$ and speaker identity $\boldsymbol{U}$ are in one-to-one correspondence. That is, for a speaker $\boldsymbol{u}$ who produced speech $\boldsymbol{x}$, $P(\boldsymbol{U} = \boldsymbol{u}|\boldsymbol{S_x}) = 1$ and for other speakers $\boldsymbol{v}(\boldsymbol{v} \neq \boldsymbol{u})$ , $P(\boldsymbol{U} = \boldsymbol{v}|\boldsymbol{S_x}) = 0$ .*

### 3.1 PROBLEM FORMULATION

Given a dataset of multi speakers and their audio recordings $\mathcal{X}$ , where speaker $u$ has $N_u$ audio recordings. Formally, for each speaker $u$, $\mathcal{X}_u = \{\boldsymbol{x_{u_i}}\}_{i=1}^{N_u}$. For every input speech $\boldsymbol{x} \in \mathcal{X}$, we use $T$ to represent the number of frames of speech $\boldsymbol{x}$, noted that $T$ is constant, which means that

for any speech segments longer than $T$, we randomly select $T$ frames, at the same time, for those speech segments with a length shorter than $T$, we pad them with constant. In this case, $x$ can also expressed as $x(1:T)$ which is a random process randomly sampled from the speech distribution $p_X(\cdot|C_X = C_x, S_X = S_x)$. Here, we assume that every speech $x$ can be expressed as a function $f(\cdot)$ of content embedding $C_x$ and speaker embedding $S_x$. That is, $x = f(C_x, S_x)$. And, we also assume that $x$ is uniquely determined by $C_x$ and $S_x$. Formally, $x_1 = x_2$ only if $C_{x_1} = C_{x_2}$ and $S_{x_1} = S_{x_2}$. Based on this assumption, for $\forall\, x_i, x_j \in \mathcal{X}$, the following formula must be true:

$$\mathcal{I}(S_{x_i}, f(C_x, S_{x_i})) = \mathcal{I}(S_{x_j}, f(C_x, S_{x_j})), \quad \mathcal{I}(C_{x_i}, f(C_{x_i}, S_x)) = \mathcal{I}(C_{x_j}, f(C_{x_j}, S_x)). \tag{3}$$

Now, assume two speech $x_1$ and $x_2$ from speaker $u$ and speaker $v$ respectively, $x_1 = f(C_{x_1}, S_{x_1})$ and $x_2 = f(C_{x_2}, S_{x_2})$. Our goal is to design a speech conversion framework to generate a new speech $\widehat{x}_{1\to 2}$ which preserves the content information of $x_1$ but the speaker information is matched with $x_2$. Formally, an ideal converted speech should satisfy the following forms:

$$\mathcal{I}(S_{x_1}, x_1) = \mathcal{I}(S_{x_2}, \widehat{x}_{1\to 2}), \qquad \mathcal{I}(C_{x_2}, x_2) = \mathcal{I}(C_{x_1}, \widehat{x}_{1\to 2}). \tag{4}$$

Based on the above assumption, the formula in Eq. (4) can be equivalent to $\widehat{x}_{1\to 2} = f(C_{x_1}, S_{x_2})$. This conclusion is quite strong, and the formal proof will be presented in the appendix.

## 3.2 THE PROPOSED FRAMEWORK

In this section, we will introduce the core component of our proposed framework. As is illustrated in Figure 1(a), our framework contains four modules. The first module is an encoder $E$, which learns a latent variable $Z$ from input speech $x$. Here we expect $Z$ to be a specific function of content embedding $C_x$ and speaker embedding $S_x$. Which can be formulated as:

$$Z = E(x) = E(f(C_x, S_x)) = S_x \oplus C_x \tag{5}$$

where $\oplus$ means concat operation. In this case, $Z$ contains both linguistic information and timbre information and when we divide $Z$ into two parts, the first part is the estimated speaker embedding $S_x$ while the second part is the estimated content embedding $C_x$.

However, without any constraints, the encoder may learn meaningless speech representation. To address this problem, two different types of classifiers are used to encourage our encoder to output the target $Z$. Specifically, when we divide $Z$ into two parts, the first part is put into a common speaker classifier $C_1$ to identify the speaker as correctly as possible. At the same time, the latter part is put into an adversarial classifier $C_2$. Noted that $C_2$ also expects to correctly recognize the speaker's identity like $C_1$, the only difference is that a **GRL** is included in $C_2$, which will make the encoder expect to fool the adversarial classifier so that it cannot classify correctly. This two different classifiers will finally converge when the first part of $Z$ is very closely related with the speaker information and the second part of $Z$ is sufficiently speaker-independent such that the adversarial classifier $C_2$ is not able to identify the speaker. Then we regard the first part of $Z$ as estimated speaker embedding $S_x$ and the second part of $Z$ is regarded as estimated content embedding $C_x$.

The last module in our framework is a decoder $D$, which will output a reconstruct speech $\widehat{x}$ from input latent variable $Z'$. It is worth noting that here the latent variable $Z'$ is not same with $Z$ since $Z'$ is produced by concating with $C_x$ and the vector norm of $S_x$ rather than original speaker embedding $S_x$.

## 3.3 HOW AND WHY DOES IT WORK

Here we will discuss how and why our model can induce style embedding and content embedding into independent representation spaces simultaneously.

In training phase, a speech $x_u$ produced by a speaker $u$ is selected to feed into the encoder $E$ to output a latent variable $Z_u = E(x_u, \theta_e)$, where $\theta_e$ are the parameters of the encoder. After that, the output of encoder is then divided into two parts, the vector norm of the first part of $Z_u$ and the speaker identity $u$ are fed into a common speaker classifier $C_1$ while the estimated content embedding $C_{x_u}$ which is the second part of $Z_u$ and the speaker identity $u$ are fed into an adversarial speaker classifier $C_2$ to predict the probability of the speaker identity according to estimated speaker

embedding $S_{x_u}$ and content embedding $C_{x_u}$ respectively, i.e. $P_u = C_1(S_{x_u}, u, \theta_{c_1})$, $P'_u = C_2(C_{x_u}, u, \theta_{c_2})$ . Where $\theta_{c_1}$ and $\theta_{c_2}$ denote the parameters of common speaker classifier and adversarial speaker classifier respectively. Subsequently, new latent variable $Z'_u$ is then fed into decoder $D$ to output a reconstruct speech $x'_u = D(Z'_u, \theta_d)$. Where $\theta_d$ refer to the parameters of decoder.

Our goal is to disentangle the content embedding $C_x$ and speaker embedding $S_x$ from a speech $x$. Formally, the ideal content embedding and speaker embedding should satisfy $\mathcal{I}(C_x, S_x) = 0$ . In addition, since the process from a speaker $u$ to his speech $x_u$ to speaker embedding $S_{x_u}$ is a MarKov Chain, we can say that $\mathcal{I}(C_{x_u}, S_{x_u}) \geq \mathcal{I}(C_{x_u}, x_u) \geq \mathcal{I}(C_{x_u}, u)$ based on the MI-data processing inequality Cover (1999) . So, we use $\mathcal{I}(C_{x_u}, u)$ instead of $\mathcal{I}(C_{x_u}, S_{x_u})$ . according to the illustration of **Thm3.1**, for the ideal content embedding, $\mathcal{I}(C_{x_u}, u) = 0$. So our goal is to minimize $\mathcal{I}(C_{x_u}, u)$ and we use the adversarial-classification loss function to minimize the upper bound. At the same time, the common-classification loss function was designed to encourage the speaker embedding $S_{x_u}$ to be as closely related to speaker identity $u$ as possible. They can be expressed as:

$$\mathcal{L}_{\text{com-cls}}(\boldsymbol{\theta_e}, \boldsymbol{\theta_{c_1}}) = -\sum_{k=1}^{K} I(u == k) \log p_k \tag{6}$$

$$\mathcal{L}_{\text{adv-cls}}(\boldsymbol{\theta_e}, \boldsymbol{\theta_{c_2}}) = -\sum_{k=1}^{K} I(u == k) \log p'_k \tag{7}$$

Where $I(\cdot)$ is the indicator function, $K$ is the number of speakers and $u$ denotes speaker who produced speech $x$, and $p_k$ is the probability of speaker $k$ . During training, for $\mathcal{L}_{\text{com-cls}}$ , $\boldsymbol{\theta_e}$ and $\boldsymbol{\theta_{c_1}}$ are all optimized to minimize the classification loss to better identify the corresponding speaker. But for $\mathcal{L}_{\text{adv-cls}}$ , $\boldsymbol{\theta_{c_2}}$ are still optimized to minimize the classification loss, whereas $\boldsymbol{\theta_e}$ are optimized to maximize the classification loss to fool the classifier. Ideally, under these two constraints, the first part of the output of encoder will be more closely related to speaker information while the second part of the output of encoder will be sufficiently speaker-independent so that the classifier can not identify the speaker. And then, we can easily get the ideal content embedding and speaker embedding at the same time.

Unfortunately, the truth is, when there are only the above two classification constraints, three different results may occur, as is illustrated in Figure 2, when estimated speaker embedding $S_x$ contain some content information, like Figure 2(a), correspondingly, for estimated content embedding $C_x$, these content information will be lost. In this case, the estimated content embedding $S_x$ are still speaker-independent and the common classifier can still correctly identify the speaker from estimated speaker embedding. The only drawback is that some content information will eventually be lost because we replace $S_x$ with it's vector norm $\widehat{S}_x$, which will lead to imperfect reconstruction. That is why we propose the reconstruct-loss function between input speech $x$ and the reconstruct speech $x'$ . That is, $\mathcal{L}_{\text{recon}} = \mathbb{E}[\|x' - x\|_2^2]$. The key logic behind setting this loss function is clear: to minimize the reconstruction loss, the estimated content embedding $C_x$ will be encouraged to carry all the content information to avoid the results of Figure 2(a).

Or, consider such a case, the estimated content embedding $C_x$ may contain some speaker information that can not be completely recognized by the adversarial speaker classifier $C_2$. When it happens, both the common classifier $C_1$ and the adversarial classifier $C_2$ can still work well, and even our model can perform well in the speech reconstruction task. In other words, the above three objective functions can not restrict the occurrence of such a situation as Figure 2(b). But, in this case, the decoder may ignore $S_x$ because the estimated content embedding $C_x$ has provided both content information and speaker information. As a result, our model can only reconstruct speech but can not convert any speech. To address this problem, we provide the code-reconstruction loss function between the latent variable $Z$ and reconstruct latent variable $\widehat{Z}$ which is produced from the reconstruct speech $x'$ , $\mathcal{L}_{\text{code-recon}} = \mathbb{E}[\|\widehat{Z} - Z\|_2^2]$ . This objective function will constrain the decoder to focus on the estimated speaker embedding $S_x$. At the same time, since the speaker information is provided by the estimated speaker embedding, $S_x$ were also encouraged to carry all the speaker information to minimize $\mathcal{L}_{\text{code-recon}}$ . The full objective function can be computed as:

$$L(\boldsymbol{\theta_e}, \boldsymbol{\theta_d}) = \mathcal{L}_{\text{recon}} + \alpha \mathcal{L}_{\text{code-recon}} + \beta \mathcal{L}_{\text{com-cls}} - \lambda \mathcal{L}_{\text{adv-cls}} \tag{8}$$

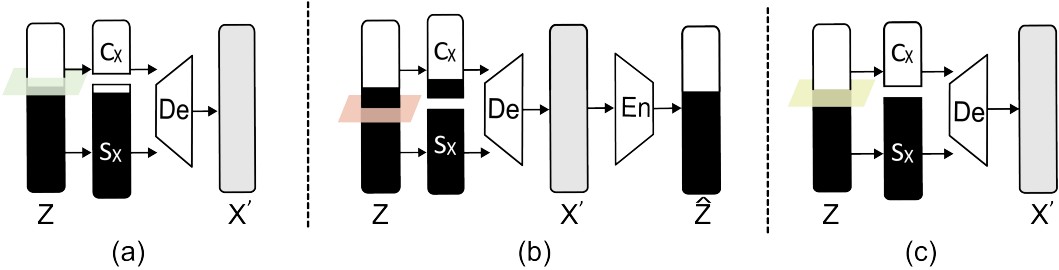

Figure 2: Three different case of disentangle different parts. Here $\boldsymbol{Z}$ are the latent variable output from the input speech, $\boldsymbol{C_x}$ denote the estimated content embedding and $\boldsymbol{S_x}$ refer to the estimated speaker embedding, both $\boldsymbol{C_x}$ and $\boldsymbol{S_x}$ are a part of the latent variable $\boldsymbol{Z}$, in addition, $\boldsymbol{x'}$ is the reconstruct speech and $\widehat{\boldsymbol{z}}$ are regarded as reconstruct latent variable. $\widehat{\boldsymbol{z}} = E(\boldsymbol{x'})$ . Whether $\boldsymbol{C_x}$ and $\boldsymbol{S_x}$ are properly selected may lead to the above three different situations

Where $\alpha$ , $\beta$, and $\lambda$ refers to the weight of $\mathcal{L}_{\text{code-recon}}$, $\mathcal{L}_{\text{com-cls}}$, and $\mathcal{L}_{\text{adv-cls}}$ respectively. With this objective loss function, an ideal case as Figure 2(c) will eventually appear. Specifically, in this case, two important assumptions will be met.

- The estimated content embedding $\boldsymbol{C_x}$ carry all content information but not contain any speaker information. On the contrary, the estimated speaker embedding $\boldsymbol{S_x}$ contain all speaker information but have no content information. In other words, now $\boldsymbol{C_x}$ are the real content embedding and $\boldsymbol{S_x}$ are the real speaker embedding. So that for any input speech $\boldsymbol{x}$, there must be $\boldsymbol{x} = f(\boldsymbol{C_x}, \boldsymbol{S_x}) \sim p_X(\cdot | \boldsymbol{C_X} = \boldsymbol{C_x}, \boldsymbol{S_X} = \boldsymbol{S_x})$ .

- With $\boldsymbol{C_x}$ and $\boldsymbol{S_x}$, an perfect reconstruction is achieved. i.e. $\boldsymbol{x} = \boldsymbol{x'}$ . Where $\boldsymbol{x'}$ is the reconstruction of $\boldsymbol{x}$ through the likelihood $p_{\theta_d}(\boldsymbol{x} | \boldsymbol{C_x}, \boldsymbol{S_x})$ .

Futher, based on the above two conditions and our previous assumptions, we then say for any index $\boldsymbol{i}$, there must be $\boldsymbol{x_i} = f(\boldsymbol{C_{x_i}}, \boldsymbol{S_{x_i}}) = \boldsymbol{x_i'} \sim p_{x_i'}(\cdot | \boldsymbol{C_{x_i}}, \boldsymbol{S_{x_i}})$ . That is, in the process of VC, if we define source utterance $\boldsymbol{x_1} = f(\boldsymbol{C_{x_1}}, \boldsymbol{S_{x_1}})$, target utterance $\boldsymbol{x_2} = f(\boldsymbol{C_{x_1}}, \boldsymbol{S_{x_2}})$, and $\boldsymbol{x_{1 \to 2}}$ is regarded as the converted speech. Then We can finally find the best encoder parameters $\boldsymbol{\theta_e}$ and decoder parameters $\boldsymbol{\theta_d}$ to satisfy the following formula: $L(\boldsymbol{\theta_e}, \boldsymbol{\theta_d}) = -\lambda \mathcal{L}_{\text{adv-cls}} = -\lambda log \frac{1}{K}$ and

$$KL(p_{X_{1 \to 2}}(\cdot | \boldsymbol{C_{x_1}}, \boldsymbol{S_{x_2}}) \| p_X(\cdot | \boldsymbol{C_X} = \boldsymbol{C_{x_1}}, \boldsymbol{S_X} = \boldsymbol{S_{x_2}})) = 0 \qquad (9)$$

where $\boldsymbol{K}$ is the number of speakers, $KL(\cdot \| \cdot)$ is the KL-divergence.

Which means the optimizer of loss function in Eq.(8) would imperceptibly satisfy the ideal speech conversion in Eq.(4). That's why we say our method can achieve distribution-matching style transfer as GAN and AutoVC.

## 4 EXPERIMENTS

In this section, we will evaluate the performance of our ClsVC on traditional many-to-many VC tasks and one-shot VC tasks. Specifically, many-to-many VC means that in conversion phase, both the selected source speaker and the target speaker are all appeared in training process. In contrast, one-shot VC focuses on some more difficult tasks, in which both the source speaker and target speaker are unseen in training dataset, and only one utterance of source and target speakers are required. Besides, we also empirically validate the convenience and robustness of the ClsVC framework. We will present the audio demo and further details may be found in our implementation code.

### 4.1 DATASETS AND CONFIGURATIONS

To evaluate the performance of different model in VC tasks, comparative experiments are conducted on VCTK Veaux et al. (2016), a high-fidelity multi-speaker English speech corpus. This corpus

contains 46 hours of speech data produced by 109 English speakers from different countries. In our work, the entire dataset is randomly divided into 2 sets: 100 speaker's recordings for training and other 9 speaker's recordings for testing. Before training, the sampling rate of all recordings is re-sampled to 16KHz, and the mel-spectrograms are computed through a short-time Fourier transform (STFT) with Hann windowing, where 1024 for FFT size, 1024 for window size and 256 for hop size. The STFT magnitude is transformed to the mel scale using 80 channel mel filter bank spanning 90 Hz to 7.6 kHz.

ClsVC is trained with batch size of sixteen for 200K steps on one NVIDIA V100 GPU, using the ADAM optimizer with $\beta_1 = 0.9, \beta_2 = 0.98$. The weight in Eq.(8) are set to $\alpha = 0.1$, $\beta = 0.1$, $\lambda = 0.5$. StarGAN-VC2[1], AutoVC[2], VAE[3], and VQVC+[4] are chosen as the baseline models.

## 4.2 Comparison

To compare the performance of different models in VC tasks, we use both objective and subjective tests. Specifically, the Mel-Cepstral Distortion(MCD) between converted speech and the ground truth target speech is used as our objective evaluation to measure the similarity of the transferred voice and the real voice from the target speaker. And, we invited 12 humans (7 male and 5 female) participants to evaluate the quality of some converted speech generated from different models. After hearing every speech, the subjects are asked to choose a score from 1-5 points of the naturalness of the converted speech. The higher the score, the better the audio quality of converted speech, which we called the Mean Opinion Score(MOS) test. In addition, all participants are also asked to take Voice Similarity Score(VSS) test. Where groups of utterances are rated with a score of 1-5 on the voice similarity, in each group, there are five converted utterances generated from AutoVC, VQVC+, VAE, StarGAN-VC2 and ClsVC respectively, and one real utterance from the target speaker. In VSS test, we calculate the score according to the timbre similarity given by the tester. Specifically, the similarity score of 5 corresponds to the converted speech most similar to the ground truth speech, while the similarity score of 1 indicates that the tester does not think that the converted speech and the ground truth speech come from the same speaker. The results appear as shown below:

Table 1: Comparison of different models in traditional VC and one-shot vc

| Methods | Traditional VC | | | One-Shot VC | | |
|---|---|---|---|---|---|---|
| | MCD | MOS | VSS | MCD | MOS | VSS |
| VQVC+ | $7.08 \pm 0.22$ | $3.31 \pm 0.90$ | $3.42 \pm 0.85$ | $8.41 \pm 0.08$ | $2.75 \pm 0.84$ | $3.11 \pm 0.88$ |
| AutoVC | $4.34 \pm 0.12$ | $3.81 \pm 1.14$ | $3.45 \pm 0.76$ | $7.66 \pm 0.17$ | $2.61 \pm 0.73$ | $2.91 \pm 0.72$ |
| VAE | $5.63 \pm 0.21$ | $3.33 \pm 0.87$ | $3.07 \pm 0.84$ | — | — | — |
| StarGAN-VC2 | $6.28 \pm 0.09$ | $3.45 \pm 1.01$ | $3.59 \pm 0.87$ | — | — | — |
| **ClsVC** | $\mathbf{4.06 \pm 0.11}$ | $\mathbf{4.09 \pm 0.86}$ | $\mathbf{4.03 \pm 0.82}$ | $\mathbf{4.81 \pm 0.13}$ | $\mathbf{4.18 \pm 0.73}$ | $\mathbf{3.75 \pm 0.74}$ |

As quoted in Table 1, in the traditional VC task, compared with other baseline models, our ClsVC has achieved the best results in both subjective and objective tests, which indicates that the speech produced by our method is much better than baselines'. The results of VSS test show that compared with VQVC+, AutoVC, VAE and StarGAN-VC2, our method makes the converted speech learn better timbre information, which improves the conversion effect.

For one-shot VC task, since VAE and StarGAN-VC2 can not achieve voice conversion for unseen speakers, we just compare our method with VQVC+ and AutoVC. The result shows that with only one utterance available for unseen speakers, the performance of AutoVC is greatly reduced, previous studies have reported this phoneme Tan et al. (2021). By comparison, even for unseen speakers and only one utterance is available, our ClsVC still achieved amazing results.

It is worth noting to say that from Table 1, the objective and subjective results are not completely consistent. For example, the mean score of MCD of VAE is 5.63, which is lower than that of StarGAN-VC2, which means that the quality of speech generated from VAE should be higher than

---

[1]`https://github.com/SamuelBroughton/StarGAN-Voice-Conversion-2`
[2]`https://github.com/auspicious3000/autovc`
[3]`https://github.com/vsimkus/voice-conversion`
[4]`https://github.com/ericwudayi/SkipVQVC`

StarGAN-VC2, but the MOS results are just the opposite. This indicates that sometimes a speech with a lower score of MCD may still have inferior audio quality. To issue the problem, we add another objective experiment. By applying a well-known open-source speech detection toolkit, *Resemblyzer* (https://github.com/resemble-ai/Resemblyzer), we conduct a fake speech detection test to compare the similarity of 11 unknown utterances (6 real ones, 5 fakes which are generated from different models respectively) against ground truth reference audio. The results are shown in Figure 5.

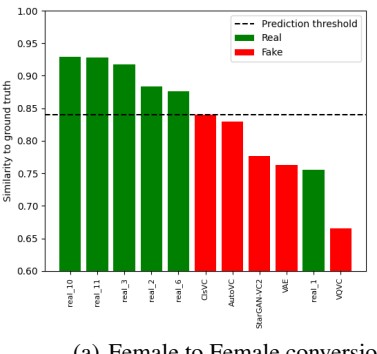 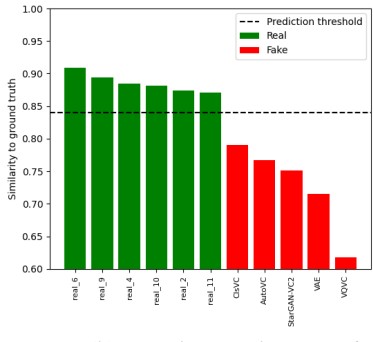

(a) Female to Female conversion         (b) Female to Male conversion

Figure 3: Results of fake speech detection. The prediction threshold in our test is 0.84

The experimental results support our conclusion, for both same-gender VC (Female to Female) and cross-gender VC (Female to Male), our ClsVC always outperforms other baseline models. All these objective and subjective experiments show that ClsVC is a state-of-the-art framework for VC under both traditional and one-shot conditions.

### 4.3 NECESSITY OF SOME LOSS FUNCTIONS

In this section, we will prove experimentally that the loss function we proposed is necessary for our framework. Specifically, we design an ablation experiment to observe the effect of $\mathcal{L}_{\text{code-recon}}$ on our framework. We retrain ClsVC without $\mathcal{L}_{\text{code-recon}}$, called 'ClsVCs'. Based on our assumption, without the constrain of the code-reconstruction loss function, the estimated speaker embedding $S_x$ may be ignored by the decoder, and our model can finally achieve a good self reconstruction but can not convert any speech. To test this hypothesis, we reuse *Resemblyzer* to do Speaker diarization test. In this test, we need prepare some ground truth speech from the source speaker and target speaker respectively, then we input a converted speech produced by 'ClsVC' and 'ClsVCs' respectively, and this toolkit will automatically identify the probability of who the input speech belongs to. For further comparison, we divide the converted audio into two gender groups, the one is same-gender conversion, including male to male and female to female VC, the other one is cross-gender conversion, including female to male and male to female VC, results of Speaker classification test are summarized in Figure 4(a).

Besides, We also select some self reconstruction speech of the source speaker to compare the performance of 'ClsVC' and 'ClsVCs' in the speech reconstruction task. In practice, 12 human participants were invited to hear a real speech and two reconstructed speech produced by 'ClsVC' and 'ClsVCs' respectively, they need to evaluate the similarity between the reconstructed speech and real speech and choose one that was more similar to the ground truth. In addition, if it is difficult to judge, they can also choose the 'Fair' option. We draw the result into the following Figure 4(b) .

The Results show that in the speech reconstruction task, 'ClsVCs' performs slightly worse than our 'ClsVC', but in the VC task, the performance of 'ClsVCs' is quite poor. As shown in Figure 4(a), the converted speech produced by 'ClsVCs' still has high similarity with the source speaker and low similarity with the target speaker, which indicates that ClsVCs cannot achieve VC without the constraints of $\mathcal{L}_{\text{code-recon}}$ .

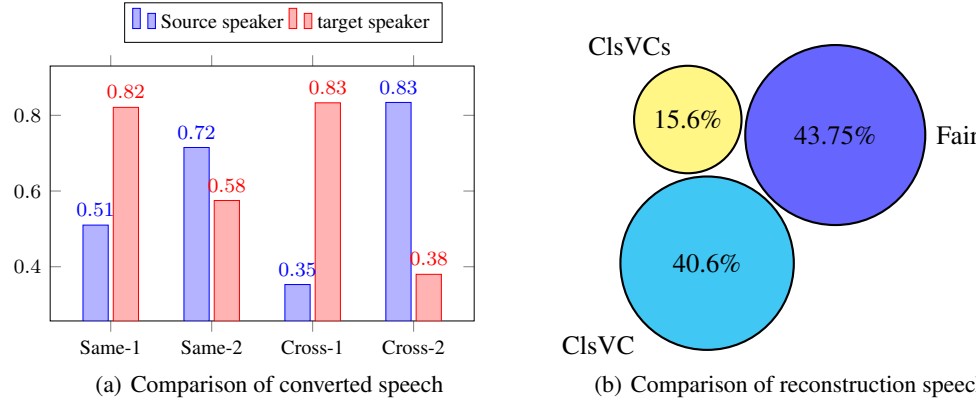

(a) Comparison of converted speech      (b) Comparison of reconstruction speech

Figure 4: Results of ablation experiment. 1:ClsVC; 2:ClsVCs; Same: same-gender conversion; Cross: cross-gender conversion;

### 4.4 DIMENSIONS OF TWO PARTS OF LATENT VARIABLE

Here we will discuss how to devide the content embedding $C_x$ and speaker embedding $S_x$ from the latent space $Z$. In AutoVC, it is important to select the size of bottleneck carefully to make the estimated content embedding contain all content information but have no timbre information. But in our model, as we discussed before, no matter how we divide it, the first part of $Z$ tends to be the ideal content embedding, and the second part tends to be the ideal speaker embedding. In this case, we can determine the dimensions of $C_x$ and $S_x$ at will and it will be very convenient for us to get content embedding and speaker embedding with only one encoder.

In order to verify that the decoupling ability of our model is equivalent under different partition modes, we retrain ClsVC by changing the dimensions of $C_x$ and $S_x$. Specifically, in ClsVC, the dimensions of $C_x$ and $S_x$ are 32 and 256, respectively. Now we retrain ClsVC by changing them to 32 and 64, called 'ClsVC-dim1', or, to 64 and 32, clled 'ClsVC-dim2'. In addition, we trained another 'clsvc-dim3' with both the dimensions of $C_x$ and $S_x$ in this model are 64. We select some unseen speakers' utterances(100 utterances per speaker) to these models to obtain the estimated speaker embedding $S_x$, then we plotted $S_x$ in 2-D space with t-SNE in Figure 5.

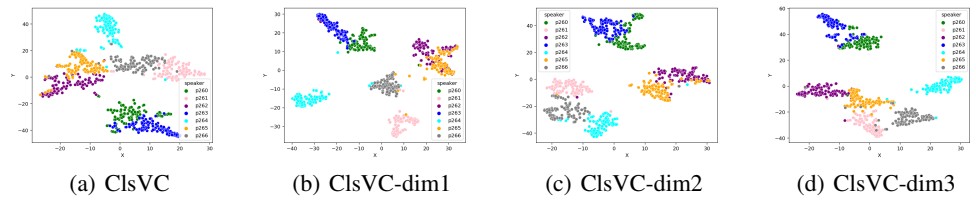

(a) ClsVC      (b) ClsVC-dim1      (c) ClsVC-dim2      (d) ClsVC-dim3

Figure 5: The visualization of speaker embedding. None of these speaker appeared in training.

Results shown in Figure 5 indicate that the content and timbre information will always be well separated even we change the division proportion of latent space.

## 5 CONCLUSION

In this paper, we proposed ClsVC, a novel VC system learning latent speech representation. During training, a common speaker classifier is proposed to encourage the estimated speaker embedding to become more and more related to the speaker identity and an adversarial classifier will focus the estimated content embedding more speaker-independent. We also introduce other objective functions to make the encoder learn the ideal latent space. All subjective and objective experimental results show that the method we proposed is state-of-the-art.

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

## 6 APPENDIX

Here we will present the formal proof of some equations which are appeared in the text of this paper.

About Eq.(3):

Given $\mathcal{I}(X, Y) = \int_X \int_Y P(X, Y) \log \frac{P(X,Y)}{P(X)P(Y)}$ , we can infer that

$$\mathcal{I}(\boldsymbol{S}_{\boldsymbol{x}_i}, f(\boldsymbol{C}_{\boldsymbol{x}}, \boldsymbol{S}_{\boldsymbol{x}_i})) = \int_{\boldsymbol{S}_{\boldsymbol{x}_i}} \int_{f(\boldsymbol{C}_{\boldsymbol{x}}, \boldsymbol{S}_{\boldsymbol{x}_i})} P(\boldsymbol{S}_{\boldsymbol{x}_i}, f(\boldsymbol{C}_{\boldsymbol{x}}, \boldsymbol{S}_{\boldsymbol{x}_i})) \log \frac{P(\boldsymbol{S}_{\boldsymbol{x}_i}, f(\boldsymbol{C}_{\boldsymbol{x}}, \boldsymbol{S}_{\boldsymbol{x}_i}))}{P(\boldsymbol{S}_{\boldsymbol{x}_i})P(f(\boldsymbol{C}_{\boldsymbol{x}}, \boldsymbol{S}_{\boldsymbol{x}_i}))}$$

Where

$$\frac{P(\boldsymbol{S}_{\boldsymbol{x}_i}, f(\boldsymbol{C}_{\boldsymbol{x}}, \boldsymbol{S}_{\boldsymbol{x}_i}))}{P(\boldsymbol{S}_{\boldsymbol{x}_i})P(f(\boldsymbol{C}_{\boldsymbol{x}}, \boldsymbol{S}_{\boldsymbol{x}_i}))} = \frac{P(\boldsymbol{S}_{\boldsymbol{X}} = \boldsymbol{S}_{\boldsymbol{x}_i})P(f(\boldsymbol{C}_{\boldsymbol{X}}, \boldsymbol{S}_{\boldsymbol{X}}) = f(\boldsymbol{C}_{\boldsymbol{x}}, \boldsymbol{S}_{\boldsymbol{x}_i})|\boldsymbol{S}_{\boldsymbol{X}} = \boldsymbol{S}_{\boldsymbol{x}_i})}{P(\boldsymbol{S}_{\boldsymbol{X}} = \boldsymbol{S}_{\boldsymbol{x}_i})P(f(\boldsymbol{C}_{\boldsymbol{X}}, \boldsymbol{S}_{\boldsymbol{X}}) = f(\boldsymbol{C}_{\boldsymbol{x}}, \boldsymbol{S}_{\boldsymbol{x}_i}))}$$

$$= \frac{P(f(\boldsymbol{C}_{\boldsymbol{X}}, \boldsymbol{S}_{\boldsymbol{X}}) = f(\boldsymbol{C}_{\boldsymbol{x}}, \boldsymbol{S}_{\boldsymbol{x}_i})|\boldsymbol{S}_{\boldsymbol{X}} = \boldsymbol{S}_{\boldsymbol{x}_i})}{P(f(\boldsymbol{C}_{\boldsymbol{X}}, \boldsymbol{S}_{\boldsymbol{X}}) = f(\boldsymbol{C}_{\boldsymbol{x}}, \boldsymbol{S}_{\boldsymbol{x}_i}))}$$

Note that we have assumed that the value of $f(\cdot)$ is unique. It is easily to refer that the probability of $f(C_{\boldsymbol{X}}, \boldsymbol{S}_{\boldsymbol{X}}) = f(C_{\boldsymbol{x}}, \boldsymbol{S}_{\boldsymbol{x}_i})$ is equivalent to $C_{\boldsymbol{X}} = C_{\boldsymbol{x}}, \boldsymbol{S}_{\boldsymbol{X}} = \boldsymbol{S}_{\boldsymbol{x}_i}$, so that

$$P(f(C_{\boldsymbol{X}}, \boldsymbol{S}_{\boldsymbol{X}}) = f(C_{\boldsymbol{x}}, \boldsymbol{S}_{\boldsymbol{x}_i})) = P(C_{\boldsymbol{X}} = C_{\boldsymbol{x}}, \boldsymbol{S}_{\boldsymbol{X}} = \boldsymbol{S}_{\boldsymbol{x}_i})$$

$$P(f(C_{\boldsymbol{X}}, \boldsymbol{S}_{\boldsymbol{X}}) = f(C_{\boldsymbol{x}}, \boldsymbol{S}_{\boldsymbol{x}_i})|\boldsymbol{S}_{\boldsymbol{X}} = \boldsymbol{S}_{\boldsymbol{x}_i}) = P(C_{\boldsymbol{X}} = C_{\boldsymbol{x}}, \boldsymbol{S}_{\boldsymbol{X}} = \boldsymbol{S}_{\boldsymbol{x}_i}|\boldsymbol{S}_{\boldsymbol{X}} = \boldsymbol{S}_{\boldsymbol{x}_i}) = P(C_{\boldsymbol{X}} = C_{\boldsymbol{x}})$$

In this case

$$\frac{P(\boldsymbol{S}_{\boldsymbol{x}_i}, f(C_{\boldsymbol{x}}, \boldsymbol{S}_{\boldsymbol{x}_i}))}{P(\boldsymbol{S}_{\boldsymbol{x}_i})P(f(C_{\boldsymbol{x}}, \boldsymbol{S}_{\boldsymbol{x}_i}))} = \frac{P(C_{\boldsymbol{X}} = C_{\boldsymbol{x}})}{P(C_{\boldsymbol{X}} = C_{\boldsymbol{x}}, \boldsymbol{S}_{\boldsymbol{X}} = \boldsymbol{S}_{\boldsymbol{x}_i})}$$

So the following formula must be true

$$\mathcal{I}(\boldsymbol{S}_{\boldsymbol{x}_i}, f(C_{\boldsymbol{x}}, \boldsymbol{S}_{\boldsymbol{x}_i})) = \int_{\boldsymbol{S}_{\boldsymbol{x}_i}} \int_{f(C_{\boldsymbol{x}}, \boldsymbol{S}_{\boldsymbol{x}_i})} P(C_{\boldsymbol{X}} = C_{\boldsymbol{x}}, \boldsymbol{S}_{\boldsymbol{X}} = \boldsymbol{S}_{\boldsymbol{x}_i}) \log \frac{P(C_{\boldsymbol{X}} = C_{\boldsymbol{x}})}{P(C_{\boldsymbol{X}} = C_{\boldsymbol{x}}, \boldsymbol{S}_{\boldsymbol{X}} = \boldsymbol{S}_{\boldsymbol{x}_i})}$$

according to the conclusion of **Thm.3.1** and **Thm.3.2**, we can say that $P(C_{\boldsymbol{X}} = C_{\boldsymbol{x}}, \boldsymbol{S}_{\boldsymbol{X}} = \boldsymbol{S}_{\boldsymbol{x}_i}) = P(C_{\boldsymbol{X}} = C_{\boldsymbol{x}}, \boldsymbol{S}_{\boldsymbol{X}} = \boldsymbol{S}_{\boldsymbol{x}_j})$ for any index $i$ and $j$. So there must be

$$\mathcal{I}(\boldsymbol{S}_{\boldsymbol{x}_i}, f(C_{\boldsymbol{x}}, \boldsymbol{S}_{\boldsymbol{x}_i})) = \mathcal{I}(\boldsymbol{S}_{\boldsymbol{x}_j}, f(C_{\boldsymbol{x}}, \boldsymbol{S}_{\boldsymbol{x}_j}))$$

Similarly

$$\mathcal{I}(C_{\boldsymbol{x}_i}, f(C_{\boldsymbol{x}_i}, \boldsymbol{S}_{\boldsymbol{x}})) = \mathcal{I}(C_{\boldsymbol{x}_j}, f(C_{\boldsymbol{x}_j}, \boldsymbol{S}_{\boldsymbol{x}}))$$

About Eq.(4):

We have shown when $\widehat{\boldsymbol{x}}_{1 \to 2} = f(C_{\boldsymbol{x}_1}, \boldsymbol{S}_{\boldsymbol{x}_2})$, there must be

$$\mathcal{I}(\boldsymbol{S}_{\boldsymbol{x}_1}, \boldsymbol{x}_1) = \mathcal{I}(\boldsymbol{S}_{\boldsymbol{x}_1}, f(C_{\boldsymbol{x}_1}, \boldsymbol{S}_{\boldsymbol{x}_1})) = \mathcal{I}(\boldsymbol{S}_{\boldsymbol{x}_2}, f(C_{\boldsymbol{x}_1}, \boldsymbol{S}_{\boldsymbol{x}_2})) = \mathcal{I}(\boldsymbol{S}_{\boldsymbol{x}_2}, \widehat{\boldsymbol{x}}_{1 \to 2})$$
$$\mathcal{I}(C_{\boldsymbol{x}_2}, \boldsymbol{x}_2) = \mathcal{I}(C_{\boldsymbol{x}_2}, f(C_{\boldsymbol{x}_2}, \boldsymbol{S}_{\boldsymbol{x}_2})) = \mathcal{I}(C_{\boldsymbol{x}_1}, f(C_{\boldsymbol{x}_1}, \boldsymbol{S}_{\boldsymbol{x}_2})) = \mathcal{I}(C_{\boldsymbol{x}_1}, \widehat{\boldsymbol{x}}_{1 \to 2})$$

Now we assume that $\widehat{\boldsymbol{x}}_{1 \to 2} = f(C_{\boldsymbol{x}_1}, \boldsymbol{S}_{\boldsymbol{x}_j})$ where $\boldsymbol{S}_{\boldsymbol{x}_j} \neq \boldsymbol{S}_{\boldsymbol{x}_2}$. In this case

$$\mathcal{I}(\boldsymbol{S}_{\boldsymbol{x}_2}, \widehat{\boldsymbol{x}}_{1 \to 2}) = \mathcal{I}(\boldsymbol{S}_{\boldsymbol{x}_2}, f(C_{\boldsymbol{x}_1}, \boldsymbol{S}_{\boldsymbol{x}_j})) = \int_{\boldsymbol{S}_{\boldsymbol{x}_2}} \int_{f(C_{\boldsymbol{x}_1}, \boldsymbol{S}_{\boldsymbol{x}_j})} P(\boldsymbol{S}_{\boldsymbol{x}_2}, f(C_{\boldsymbol{x}_1}, \boldsymbol{S}_{\boldsymbol{x}_j})) \log \frac{P(\boldsymbol{S}_{\boldsymbol{x}_2}, f(C_{\boldsymbol{x}_1}, \boldsymbol{S}_{\boldsymbol{x}_j}))}{P(\boldsymbol{S}_{\boldsymbol{x}_2})P(f(C_{\boldsymbol{x}_1}, \boldsymbol{S}_{\boldsymbol{x}_j}))}$$

Where

$$\frac{P(\boldsymbol{S}_{\boldsymbol{x}_2}, f(C_{\boldsymbol{x}_1}, \boldsymbol{S}_{\boldsymbol{x}_j}))}{P(\boldsymbol{S}_{\boldsymbol{x}_2})P(f(C_{\boldsymbol{x}_1}, \boldsymbol{S}_{\boldsymbol{x}_j}))} = \frac{P(\boldsymbol{S}_{\boldsymbol{X}} = \boldsymbol{S}_{\boldsymbol{x}_2})P(f(C_{\boldsymbol{X}}, \boldsymbol{S}_{\boldsymbol{X}}) = f(C_{\boldsymbol{x}_1}, \boldsymbol{S}_{\boldsymbol{x}_j})|\boldsymbol{S}_{\boldsymbol{X}} = \boldsymbol{S}_{\boldsymbol{x}_2})}{P(\boldsymbol{S}_{\boldsymbol{X}} = \boldsymbol{S}_{\boldsymbol{x}_2})P(f(C_{\boldsymbol{X}}, \boldsymbol{S}_{\boldsymbol{X}}) = f(C_{\boldsymbol{x}_1}, \boldsymbol{S}_{\boldsymbol{x}_j}))}$$

Since the probability of $f(C_{\boldsymbol{X}}, \boldsymbol{S}_{\boldsymbol{X}}) = f(C_{\boldsymbol{x}_1}, \boldsymbol{S}_{\boldsymbol{x}_j})|\boldsymbol{S}_{\boldsymbol{X}} = \boldsymbol{S}_{\boldsymbol{x}_2}$ is zero, We conclude that

$$\frac{P(\boldsymbol{S}_{\boldsymbol{x}_2}, f(C_{\boldsymbol{x}_1}, \boldsymbol{S}_{\boldsymbol{x}_j}))}{P(\boldsymbol{S}_{\boldsymbol{x}_2})P(f(C_{\boldsymbol{x}_1}, \boldsymbol{S}_{\boldsymbol{x}_j}))} = 0 \quad while \quad \frac{P(\boldsymbol{S}_{\boldsymbol{x}_2}, f(C_{\boldsymbol{x}_1}, \boldsymbol{S}_{\boldsymbol{x}_2}))}{P(\boldsymbol{S}_{\boldsymbol{x}_2})P(f(C_{\boldsymbol{x}_2}, \boldsymbol{S}_{\boldsymbol{x}_2}))} = \frac{P(C_{\boldsymbol{X}} = C_{\boldsymbol{x}_1}, \boldsymbol{S}_{\boldsymbol{X}} = \boldsymbol{S}_{\boldsymbol{x}_2})}{P(C_{\boldsymbol{X}} = C_{\boldsymbol{x}_1})}$$

Similarly, if we assume that $\widehat{\boldsymbol{x}}_{1 \to 2} = f(C_{\boldsymbol{x}_i}, \boldsymbol{S}_{\boldsymbol{x}_2})$ where $C_{\boldsymbol{x}_i} \neq C_{\boldsymbol{x}_1}$. In this case

$$\mathcal{I}(C_{\boldsymbol{x}_1}, \widehat{\boldsymbol{x}}_{1 \to 2}) = \mathcal{I}(C_{\boldsymbol{x}_1}, f(C_{\boldsymbol{x}_i}, \boldsymbol{S}_{\boldsymbol{x}_2})) = \int_{C_{\boldsymbol{x}_1}} \int_{f(C_{\boldsymbol{x}_i}, \boldsymbol{S}_{\boldsymbol{x}_2})} P(C_{\boldsymbol{x}_1}, f(C_{\boldsymbol{x}_i}, \boldsymbol{S}_{\boldsymbol{x}_2})) \log \frac{P(C_{\boldsymbol{x}_1}, f(C_{\boldsymbol{x}_i}, \boldsymbol{S}_{\boldsymbol{x}_2}))}{P(C_{\boldsymbol{x}_1})P(f(C_{\boldsymbol{x}_i}, \boldsymbol{S}_{\boldsymbol{x}_2}))}$$

Where

$$\frac{P(C_{x_1}, f(C_{x_i}, S_{x_2}))}{P(C_{x_1})P(f(C_{x_i}, S_{x_2}))} = \frac{P(C_X = C_{x_1})P(f(C_X, S_X) = f(C_{x_i}, S_{x_2})|C_X = C_{x_1})}{P(C_X = C_{x_1})P(f(C_X, S_X) = f(C_{x_i}, S_{x_2}))}$$

Then we can get the same conclusion, That is

$$\frac{P(C_{x_1}, f(C_{x_i}, S_{x_2}))}{P(C_{x_1})P(f(C_{x_i}, S_{x_2}))} = 0 \quad while \quad \frac{P(C_{x_1}, f(C_{x_1}, S_{x_2}))}{P(S_{x_2})P(f(C_{x_2}, S_{x_2}))} = \frac{P(C_X = C_{x_1}, S_X = S_{x_2})}{P(S_X = S_{x_2})}$$

Which indicates that the formula in Eq.(4) is equivalent to $\widehat{x}_{1\to 2} = f(C_{x_1}, S_{x_2})$

