# OpenReview forum: "ClsVC: Learning Speech Representations with two different classification tasks."
_ICLR.cc/2022/Conference — ICLR 2022 Submitted_

### Official Review · Reviewer_DgA7 · 2021-10-31

**Correctness:** 3
**Technical Novelty And Significance:** 3
**Empirical Novelty And Significance:** 2
**Recommendation:** 5
**Confidence:** 3

**Main Review:**

Strength
1. The problem formulation is subtle and interpretative, given the premise theory constraints. The formal proof is also providing bonus points.
2. The model is compared with open-sourced widely-acknowledge frameworks and outperformed them reliably.
3. The ablation study is also a bonus point, especially analysis on latent variables.

Weakness & Questions
1. There are more than several grammatical and format-wise mistakes made in this paper. Please fix them via proof-reading, either for future publication or camera-ready version.
2. Citation is poorly formed and arranged. Please check and re-write them.
3. Problem formulation shall not be included as part of methodology. Please move them as a separate section or as part of the introduction.
4. As a researcher who has some experience in speaker recognition, I question Theorem 3.2 - Embedding and Identity naturally shall not be an objective&surjective mapping. It should only be a surjective one.
5. Does the parameters of Eq. 8 set by pilot experiments? Or from a pre-trained dataset? Not clear enough.
6. The reviewer suggests to perform some tests on publicly available datasets for VC e.g. VC2018 challenge if applicable [1]. Does not have to be extensive but need one.
7. One main weakness of this paper is during the evaluation time, multiple encoders are anyway also needed. Does not have to handle this for now but this shall be addressed and left as a future work.

[1] http://www.vc-challenge.org/vcc2018/index.html

**Summary Of The Paper:**

This paper presents a learning framework for Voice Conversion (VC) with single encoder containing both contextual and speaker information, along with some tricks on utilizing timbre and speaker-wise information in both training and inference phases. It is claimed to achieve state-of-the-art performance, outperforming most recently open-sourced models in the VC research community.

**Summary Of The Review:**

In general, the author has mixed feeling about this work. On the one hand, the proposed framework is decent with clear mathematical proof (apart from Theorem 3.2, which is not valid but essentially trivial in the whole proof). However on the other hand, more than several statements are questionable, along with wording and grammatical mistakes. Besides, the speaker embedding background information is missed, along with Theorem 3.2 which is wrong in reviewer's personal knowledge.

Therefore, the reviewer is recommend this paper to be on the margin. Even if it got accepted, significant work is expected to be done on re-phrasing, minor additional experiments, and speaker embedding related theoretical coverage.

---

### Official Review · Reviewer_fLnd · 2021-11-02

**Correctness:** 2
**Technical Novelty And Significance:** 2
**Empirical Novelty And Significance:** 2
**Recommendation:** 3
**Confidence:** 4

**Details Of Ethics Concerns:**

I did not flag anything here, but voice conversion inherently can have discrimination, privacy, and other potentially harmful issues.

**Main Review:**

The four losses have been individually proposed in the past, and the paper does not try to argue why the combination of which is novel.

The experiments in Table 1 are weak and do not show any insights. The models in Table have different architectures and different training losses. It's also unclear if these models are tuned for this particular task, on this particular data set under this particular training.

The theory developed in section 3 is weak. The first and major assumption is that linguistic content can be separated from speakers. This assumption is clearly false, because different speakers clearly have preference over how sentences are constructed and have preference over how words are pronounced. The second assumption is that the content and the speaker embedding uniquely determine the speech, or in the paper x = f(C_x, S_x). This again is false, because no single person is able to produce the exact same speech utterance twice. Another assumption due to the information processing inequality I(C_x, S_x) \geq I(C_x, x) is strong and is likely false. Based on equation (5), it really should be I(C_x, x) \geq I(C_x, E(x)) \geq I(C_x, S_x).

The approach has a few weaknesses. The first is about disentanglement. Even though adding a speaker classifier encourages pushes the speaker information to one embedding, it does not mean similar information is not in the other embedding unless a bottleneck is used. The gradient reversal layer also encourages similar effect, but again that the discriminator cannot identify speakers does not imply the speaker information is absent. Reconstructing the hidden vectors can also be also concerning, because the model can end up mapping everything to a constant vector if the weight of that loss is too strong. These weaknesses are not addressed in the paper.

The presentation has room for improvement. For example, listing the problems of other models in the introduction seems unnecessary, because the paper is not directly addressing these problems. The introduction also does not motivate the proposed approach. The background section is fragmented, jumping over multiple unrelated aspects. The theorems in section 3 should really be assumptions. There are also wording issues and typos throughout the paper. Below is a non-exhaustive list.

> However, whether they apply a bottleneck network or vector quantify technology ...

vector quantify technology --> vector quantization?

> Mutual information (MI), a crucial indicator to measure ...

This sentence does not have a verb.

> So our goal is to minimize ...

Remove "so"?

> Unfortunately, the truth is, when there are only ...

Remove "the truth is"?

> In this section, we will prove experimentally ...

prove --> show? It's possible to use experiments to refute hypotheses, but I don't think this is the intended meaning.


**Summary Of The Paper:**

The paper proposes a model for voice conversion. To achieve voice conversion, the approach is to separate linguistic content and speaker information into two embedding vectors. Voice conversion is achieved by swapping out the speaker embedding. There are four losses involved, one for reconstructing speech, one for reconstructing the embedding vectors, one for classifying speakers, and the last is a task to make speaker classification adversarially hard.

**Summary Of The Review:**

The paper does not argue why the proposed approach is novel. The experiments do not show any insight. The theory in the paper is weak, if not wrong. Several weakness of the approach are not mentioned. The presentation has a lot of room for improvement.

---

### Official Review · Reviewer_n7Fo · 2021-11-04

**Correctness:** 3
**Technical Novelty And Significance:** 2
**Empirical Novelty And Significance:** 2
**Recommendation:** 3
**Confidence:** 4

**Main Review:**

Strengths:
(1) In the proposed method, a single encoder is adopted to extract both timber (speaker) and content information from the reference speech, which simplifies the model architecture.
(2) The latent speech representation is divided into two parts, for speaker and content embeddings repressively, which provides the flexibility to change the proportion of speaker and content embeddings given the total dimension of the latent representation.
(3) Two classifiers are proposed to ensure the disentanglement of speaker and content information.  The common speaker classifier ensures the speaker embedding carries the timber information.  The adversarial speaker classifier ensures the content embedding does not contain timber information.
(4) Vector norm is applied to the learned speaker classifier to eliminate the content information from the speaker embedding.
(5) Code-reconstruction loss is proposed to force the content embedding not to contain the timber information.

Weakness:
(1) The method proposed by the paper is actually very simple and straightforward. The only difference between the proposed work with some state-of-the-art methods is the "combining" of nowadays content encoder and speaker encoder into a single encoder.  However, from the perspective view of the neural networks, the performance of a single encoder and the separate encoders should be the same.  Hence, the novelty of the proposed method is limited.
(2) There might be some errors for the technical part.  For example, for equation (8), the adv-cls loss is subtracted from the total loss.  The minimizing of the total loss will actually increase the adv-cls loss.  However, the adv-cls loss should also be minimized to optimize the parameters (θ_{c2}) of the adversarial speaker classifier.  The optimization of the encoder (θ_e) tries to "fool" the adversarial speaker classifier by the gradient reversal layer only.
(3) In addition to above (2), In the 3rd paragraph introducing code-reconstruction loss under equation 7, it is unclear how the Z_hat is derived?  If the input to the intermediate encoder (to derive Z_hat) is X', and X' is just reconstructed from the original speaker embedding (S_x) and content embedding (C_x), I do NOT think the introduction of code-reconstruction loss can solve the problem.  Or, the authors should clarify how the Z_hat (and X') is derived.
(4) In the 2nd paragraph introducing reconstruct-loss under equation 7, it is said that "In this case, the estimated content embedding S_x are still speaker-independent and the common classifier can still ...".  Here, it seems that "the estimated content embedding" should be "the estimated speaker embedding"?
(5) In the ablation study, only the effectiveness of "code-reconstruction loss" is experimented.  How about the other losses?
(6) In section 4.4, the dimension of two parts of the latent variable has been experimented to validate the flexibility of the proposed method.  The question is will the selection of the dimensions affect the performance of VC and why?
(7) The English writing of the paper should be proofread by a native speaker.  There also many topos in the paper.


**Summary Of The Paper:**

The paper has proposed a VC system called ClsVC by learning the latent speech representation from reference speech.  To ensure the disentanglement of speaker and content inforamtion, the proposed method separates the latent speech representation into two parts, for speaker and content embedding respectively.  To ensure the performance of disentanglement, a common speaker classifier and an adversarial speaker classifier are proposed.  Furthermore, a new loss called code-reconstruction loss is proposed. Experimental results validate the effectiveness of the proposed method.

**Summary Of The Review:**

Based on the above main review (especially the weaknesses of the paper), the method proposed by the paper is actually very simple and straightforward.  The novelty of the proposed method is limited.  There might also be some errors for the technical part.  The English writing of the paper needs improvement.  Hence, I do NOT think the paper should be accepted.

---

> ### Comment · Reviewer_n7Fo · 2021-11-24
> **My recommendation about the paper unchanged**
>
> Thank the authors for providing feedback about the reviews.  However as I have said in my review, the proposed method of the paper is actually quite straightforward and simple.  The novelty of the method is limited.  And the technical errors of the paper are not addressed by the feedback.  Hence, my recommendation about the paper will remain unchanged.

---

### Official Review · Reviewer_XqM2 · 2021-11-07

**Correctness:** 3
**Technical Novelty And Significance:** 2
**Empirical Novelty And Significance:** 2
**Recommendation:** 3
**Confidence:** 4

**Main Review:**

The paper proposes an approach for voice conversion. It key idea is to add classification tasks in an autoencoder approach. In particular, these classification tasks are aimed at disentangling the speaker and content representations.

1. The paper seems to be  following AutoVC,  even on the details and descriptions. Here there are no separate speaker and content encoders and additional classifiers are added on top of speaker and content speakers.


2. The paper is missing some important details. The details of the encoder and the decoder are missing. The details of the classifiers C1 and C2 are also missing.

3. Why are Theorems 3.1 and 3.2 marked as “Theorem” ? Aren’t these just the assumptions/conditions under which the framework operates ?

4.  How do we understand that the content embedding is capturing meaningful content related information ? The content embedding is trying to fool a speaker classifier, which it can do by producing any random output. The reconstruction losses can be claimed to avoid that but is there a way we can verify how much this holds empirically ? The paper talks about content and speaker information leakages into the embeddings. However, no attempt is made to study this empirically.

5. What happens when Audio X_i and Audio X_j in Fig 1(b) are utterances of the same speaker.  How well is the reconstruction in this case. If the assumptions in the paper holds with training - then the framework should be able to reconstruct the audio well.

6. Paper’s writing could be improved  substantially. Authors should proof-read the paper. There are quite a few typos as well as language errors. At times, the paper is not easy to read and understand. In several cases, the Figures (e.g 1 and 4) are not even properly discussed.

7. The classifiers take the speaker/content embeddings as well the speaker identity (\mathbf{u}) ? Is the speaker identity provided as a one-hot vector ?

8. The experiments are also not very thorough and lacking in some aspects. Moreover, some basic details are also missing. First, it's not clear how the data is partitioned for experiments. The paper says the entire dataset is divided into two sets - 100 speakers for training and 9 speakers for testing ? But the “Traditional VC” has seen speakers. So how the training and testing partitioned for those experiments ? How are all the hyperparameters selected ?

9. The MOS results have very high variance - close to 1 for all cases. This seems very high and makes it hard to interpret the performance. Also, what is the sample size for MOS study ?

10. Its not clear what Fig 4 is trying to say. It is not properly discussed.

11. For the change in dimensions, it would have been better if we could also see how the performances are affected by these changes along with the speaker embedding visualization.

12. As mentioned in point 4, it would be great if some of the claims on speaker and content embeddings are empirically studied to verify how they hold after training. No interesting insight is provided in the paper.






**Summary Of The Paper:**

This paper proposes methods for voice conversion. The proposed approach follows an encoder-decoder framework in which it adds classification tasks over the learned embeddings to enhance voice conversion. The representation space is divided into two parts - content and speaker, to enable disentangling of the content and the speaker. Experiments are shown on “traditional” and one-shot VC tasks. Objective as well as subjective scores are used to evaluate all systems.

**Summary Of The Review:**

The paper proposes a method for voice conversion which aims to use 2 classification tasks to learn content and speaker embeddings. The idea is simple and makes sense. However, the paper is not clear on several fronts. The claims are not justified properly through experiments and key details are missing as well. I think the paper in the current form is not ready for publication even though it appears to be a meaningful approach for VC.

---

> ### Author Response · Authors · 2021-11-10
> **Author response to Reviewer XqM2**
>
> Dear Reviewer XqM2:
> We are appreciated for your constructive review and address your comments below.
>
> As you said, the details for our model are not provided here due to length limitations. However, we have uploaded all the implementation codes of our model as supplementary materials. Furthermore, if necessary, we will add the details of our module in the final version.
>
> The content embedding can never produce any random output due to the reconstruction and adversarial loss constraints. The reason is simple: since a perfect reconstruction needs content information and style information, if content embedding produces random output, the speaker embedding should provide both speaker information and content information to achieve perfect reconstruction. However, as we mentioned in our paper, what we put into the decoder is the vector norm of the speaker embedding, not the initial speaker embedding. In this case, even if the content information is really included in the speaker embedding, most of the content information will be lost, and the reconstruction loss will increase. Therefore, the speaker embedding can not carry content information, and the content embedding must produce meaningful content-related information. Besides, we introduced ablation studies to show that the content embedding may contain speaker information without the code-reconstruction loss, and in this case, except for the source speaker himself, no matter who the target speaker is, the VC task cannot be completed. Fig. 4 just wants to illustrate this.
>
> When Audio X_i and Audio X_j in Fig 1(b) are utterances of the same speaker, a good reconstructed audio will be produced. Actually, you can see the result in Fig 4(b). Even without the code-reconstruction loss, the model can also achieve a good reconstruction task.
>
> About the paper writing, we are appreciated for your advice, and we will proofread the paper carefully. About Fig 4, due to the length limitations, we did not explain the results in detail. However, we explained in detail why we did this experiment, and the results (in our opinion) were noticeable.
>
> The speaker's identity is a one-dimensional label, not a one-hot vector.
>
> We are sorry for lacking details for the processing in our datasets, and we will add them. In the MOS study, every subject should listen to 12 samples produced by every method, respectively.
>
> Thanks for your advice, and we will add some experiments to show how the performances are affected by these changes along with the speaker embedding visualization.

---

### Decision · Program_Chairs · 2022-01-20

**Decision:**

Reject

**Comment:**

This paper proposes a voice conversion framework, ClsVC, which is based on disentanglement of speaker and content information in some latent space.  The authors introduce two classification constraints (a common speaker classifier and an adversarial classifier) to improve the separation of the two embeddings.  Experimental results are reported on a few voice conversion tasks with objective and subjective scores.  Reviewers have reservation about the novelty of the work which is not considered overwhelmingly significant given existing techniques. The theory and arguments on the claimed effectiveness of the disentanglement of speaker and content also raise concerns, which need to be further verified.  The experimental results need to be more convincing.  Lastly, the exposition needs significant improvements.  The authors' rebuttal answers some of the comments but a few major concerns still stand.  This paper can not be accepted given its current form.